# Feasibility and acceptability of a pilot, peer-led HIV self-testing intervention in a hyperendemic fishing community in rural Uganda

Joseph K. B. Matovu[1,2]*, Laura M. Bogart[3], Jennifer Nakabugo[1], Joseph Kagaayi[4], David Serwadda[1], Rhoda K. Wanyenze[1], Albert I. Ko[5], Ann E. Kurth[5,6]

1 Makerere University School of Public Health, Kampala, Uganda, 2 Busitema University Faculty of Health Sciences, Mbale, Uganda, 3 RAND Corporation, Santa Monica, CA, United States of America, 4 Rakai Health Sciences Program, Kalisizo, Uganda, 5 Yale School of Public Health, New Haven, CT, United States of America, 6 Yale School of Nursing, New Haven, CT, United States of America

* jmatovu@musph.ac.ug

**Data Availability Statement:** All relevant data are within the manuscript and its Supporting Information files.

## Abstract

### Background

Novel interventions are needed to reach young people and adult men with HIV services given the low HIV testing rates in these population sub-groups. We assessed the feasibility and acceptability of a peer-led oral HIV self-testing (HIVST) intervention in Kasensero, a hyperendemic fishing community (HIV prevalence: 37–41%) in Rakai, Uganda.

### Methods

This study was conducted among young people (15–24 years) and adult men (25+ years) between May and August 2019. The study entailed distribution of HIVST kits by trained "peer-leaders," who were selected from existing social networks and trained in HIVST distribution processes. Peer-leaders received up to 10 kits to distribute to eligible social network members (i.e. aged 15–24 years if young people or 25+ years if adult man, not tested in the past 3 months, and HIV-negative or of unknown HIV status at enrolment). The intervention was evaluated against the feasibility benchmark of 70% of peer-leaders distributing up to 70% of the kits that they received; and the acceptability benchmark of >80% of the respondents self-testing for HIV.

### Results

Of 298 enrolled into the study at baseline, 56.4% (n = 168) were young people (15–24 years) and 43.6% (n = 130) were adult males (25+ years). Peer-leaders received 298 kits and distributed 296 (99.3%) kits to their social network members. Of the 282 interviewed at follow-up, 98.2% (n = 277) reported that they used the HIVST kits. HIV prevalence was 7.4% (n = 21). Of the 57.1% (n = 12) first-time HIV-positives, 100% sought confirmatory HIV

**Funding:** This work was implemented as part of JKBM's Post-Doctoral Research Fellowship with a grant from the National Institutes of Health Fogarty International Center (NIH FIC D43TW010540; PIs: Riley LW, Barry M, Ko AI, Madhivanan P) and another grant from the Africa Research Excellence Fund (RF-1570024-F-MATOV). The funders had no role in study design, data collection and analysis, decision to publish, or preparation of the manuscript.

**Competing interests:** The authors have declared that no competing interests exist.

testing and nine of the ten (90%) respondents who were confirmed as HIV-positive were linked to HIV care within 1 week of HIV diagnosis.

## Conclusion

Our findings show that a social network-based, peer-led HIVST intervention in a hyperendemic fishing community is highly feasible and acceptable, and achieves high linkage to HIV care among newly diagnosed HIV-positive individuals.

## Introduction

Young people and adult men are less likely to test for HIV and to be enrolled in HIV prevention, care and treatment programs [1]. Studies show that individuals aged 15–24 years are less likely to be aware of their HIV status, to be enrolled in HIV care, and to have a suppressed viral load compared to HIV-positive persons aged 30 years or older [2, 3]. This situation is even more pronounced in fishing communities in sub-Saharan Africa, where access to HIV and other health services is usually limited due to their remote locations away from the main health facilities. In a recent paper assessing the impact of combination HIV interventions on HIV incidence in hyperendemic fishing communities in Uganda, Kagaayi et al. [4] found that linkage to HIV care among HIV-positive young men in the Kasensero fishing community increased only slightly from 3% to 28% over a six-year period (2011–2017). In another conducted in the same setting to assess HIV prevalence and uptake of HIV services among youths (15–24 Years), Mafigiri et al. [5] found a high prevalence of HIV (19.7%) amidst very low (22.4%, $n = 34$) linkage to HIV care among HIV-positive youth. When the analysis was stratified by sex, Mafigiri et al. [5] found low utilization of HIV testing and linkage to HIV care services among male youth (HIV testing: 37.3; linkage to HIV care: 6.7) compared to female youth (HIV testing: 62.7; linkage to HIV care: 28.4). These results are corroborated by Billioux et al. [3] who found that enrolment into HIV care among HIV-positive individuals aged 15–24 years in Rakai district was 28% lower than among older individuals aged 30–39 years. Inconvenient working hours for the highly mobile fisher-folk population coupled with limited access to health facilities largely account for the low HIV testing coverage rates among young people living in the fishing community [6, 7].

On the other hand, efforts to reach men with HIV testing and treatment programs continue to be hampered by hegemomonic masculinity norms [8–11]. Evidence from gender-related studies, particularly those that focus on masculinity and its effects on the uptake of facility-based health services, have found that men tend to avoid going to the health facilities because of fear that they could be presumed to be *weak* or to have HIV–which would negatively impact their "superior" social status [8, 10]. Nyamhanga et al. [8] observed that societal expectations of a 'real man' to be fearless, resilient, and emotionally stable are in direct conflict with expectations of HIV treatment programs such as agreeing to take HIV tests and disclosing one's HIV status to at least one's spouse or partner. These sentiments were also found in another study that explored men's absence from HIV treatment programs in Zimbabwe [10]. In general, men aged 25+ years are particularly missing in HIV testing and linkage to HIV care programs. Our previous study that assessed the correlates of HIV status awareness among Ugandans aged 45+ years found that only 48% had ever tested and received their HIV test results, while 23% tested and received their HIV results in the past 12 months or already knew that they are HIV positive [12]. In a nationally representative survey conducted in 2016 in

Uganda, uptake of HIV testing among men decreased with increasing age from 31.3% of men aged 25–29 years; 28.9% of men aged 30–39 years to 21.3% of men aged 40–49 years [13]. These findings generally reflect the observation that men are less represented in HIV testing and treatment programs. It is important to note that the above-mentioned findings pertain to the coverage of HIV testing among adult men in the general population but not among specific population sub-groups such as people living in high HIV prevalence fishing communities.

Given the high mobility of people in the fishing communities (fishermen tend to move along with the fish season; moving to areas with higher stocks as the fishing season wears on), it is likely that the uptake of HIV testing, and eventual linkage to HIV care, among HIV-positive fisher-folk is somewhat lower than what is reported in the general population [3]. Indeed, a study conducted among the fisher-folk at Kasenyi landing site (along the shores of Lake Victoria in Uganda) found that only 47.2% of the respondents had ever tested for HIV [14] compared to 79% in the general population [13]. These results demonstrate the need for more targeted interventions to improve both HIV testing and linkage to HIV care among the fisher-folk. Evidence from prior studies shows that interventions that offer social spaces that allow men to project non-normative masculine characteristics such as vulnerability and weakness while retaining their social status [15] and those that provide men with an opportunity to discuss HIV-related issues, including HIV testing, with fellow men [16] can improve HIV testing rates among men. However, only a few interventions have targeted men in fishing communities with interventions aimed to improve HIV testing or linkage to HIV care through men-to-men interventions [17, 18]. The objective of this study was to assess the feasibility and acceptability of a peer-led oral HIV self-testing (HIVST) intervention to improve HIV testing and linkage to HIV care among young people (15–24 years) and adult men (25+ years) in Kasensero fishing community along the shores of Lake Victoria in rural Uganda.

## Materials and methods

### Study site

Kasensero fishing community lies on the shores of Lake Victoria (closer to Uganda's southern border with Tanzania) and was among the first places where the first HIV/AIDS cases were identified in Uganda in 1982 [19]. It is made up of three study communities, namely: Kasensero landing site, Gwanda and Kyebe, arranged in the order of how close they are from the Lake shores. Our previous research in this area shows that HIV prevalence decreases as one moves away from the landing site to the hinterland [5]. Adult HIV prevalence in Kasensero fishing community is very high, ranging between 37–41% [4, 20] although HIV incidence has declined from 3.43/100py in 2011 to 1.59/100py in 2017 [4] due to increasing antiretroviral therapy (ART) coverage in the general population since 2011 [4]. However, only about 24% of young HIV-positive individuals have been linked to HIV care, largely because of the mobile nature of the fisher-folk [5, 20]. Fishing community residents are predominately male, more likely to be unmarried, highly mobile, and report higher levels of HIV-related sexual risk behaviors (including unprotected sex and alcohol use before sex) compared to residents of inland communities [21]. Kasensero fishing community is served by two health centers (Kasensero Health Center II and Kyebe Health Center III) that offer HIV testing and linkage to HIV care, among other HIV services.

### Study design and population

This feasibility and acceptability study was conducted among young people (both males and females, age 15–24 years) and adult men (25 years or older) living in Kasensero fishing community between May 6 and August 27, 2019. The primary purpose of this study was to

generate data necessary to inform the design of future interventions to assess the effect of peer-led HIV self-testing on HIV testing uptake and linkage to HIV care among young people and adult men in Ugandan fishing communities. To be eligible for study enrolment, young people and adult men had to be aged 15 years or older, last tested for HIV three or more months prior to study enrolment, HIV-negative or of unknown HIV status at the time of nomination and be nominated by a trained peer-leader in the community. The term 'peer-leader' was used to refer to a lay member of the community, selected by fellow community members, who was trained to distribute HIV self-test kits to his/her close associates who met the study inclusion criteria (see '*intervention description*' below).

## Theory of change

The design of this peer-led HIV self-testing intervention was guided by a Theory of Change that was informed by three theories of behavior change: Socio-Ecological Model [22], Information-Motivation-Behavioral Skills (IMB) Model [23] and the Theory of Reasoned Action (TRA) [24]. The Socio-Ecological Model (SEM) recognizes the interwoven relationship that exists between the individuals and their environment; that is, while individuals are responsible for instituting and maintaining lifestyle changes necessary to reduce risk and improve health, individual behavior is determined, to a large extent, by the other members of their social networks (at the interpersonal level) and at the next level by the community in which they live. Barriers to healthy behaviors are shared among the community as a whole. As these barriers are lowered or removed, behavior change becomes more achievable and sustainable [24].

The IMB model recognizes that behavior change is a function of three primary constructs: *information* and knowledge about the behavior; the individual's *motivation* to perform the behavior; and the *behavioral skills* necessary to perform the behavior [23]. The IMB model postulates that *information* that is directly relevant to the performance of health behavior (e.g. HIV self-testing can be easily done by lay individuals) and that can be easily enacted by an individual in his or her social ecology is a critical determinant of health behavior performance. Fisher and Fisher [24] reasoned that both *motivation* and *behavioral skills* serve as additional determinants of the performance of health-related behaviors, and influence whether even well-informed individuals will be inclined to accept the recommended health options (e.g. HIV self-test kits) or be capable of effectively performing the recommended health actions, e.g. use the HIVST kits to test for HIV [25].

On the other hand, the TRA contends that a person's intention to perform a behavior (e.g. intention to test for HIV) is the main predictor of whether or not they will actually perform that behavior [23]. This intention comes as a result of a belief that performing the behavior will lead to a specific outcome, e.g. get to know one's HIV test results or enroll into HIV care if HIV-positive. Behavioral intention is determined by attitudes to behavior and subjective norms. A person's attitude toward a particular behavior is influenced by their beliefs about the outcome of the behavior and their evaluation of the potential outcome (e.g., *would use of HIV self-test kits yield accurate HIV test results*?). Subject norms refer to a person's belief of what others think about the behavior (e.g. using HIVST kits) and that person's ability to comply with what others think of HIVST.

Borrowing from these theoretical constructs, we designed our Theory of Change on the assumption that: a) young people and adult men in the community will be willing to receive HIVST kits from trained lay people in their communities (i.e. peer-leaders); b) distribution of HIVST kits will address male masculinity norms that tend to inhibit men from seeking health facility-based HIV testing services; c) young people and adult men who will be trained as peer-leaders will be willing to distribute the kits to members of their social networks; d) adult men

or young people in the peer-leaders' social networks who will receive kits will use them to self-test for HIV and to ascertain their HIV status; e) adult men or young people in peer-leaders' social networks who self-test HIV-positive will be motivated to seek confirmatory HIV testing and link to HIV care if confirmed to be HIV-positive; f) HIV self-tested individuals will not engage in high-risk sexual behaviors; and g) overall, network-based, peer-led HIVST will be well accepted and result in improved HIV testing rates and linkage to HIV care with minimal social consequences.

## Intervention description

**Formative research to inform intervention development.**   In May 2019, we conducted six focus group discussions (FGDs) with 47 participants (31 men and 16 women aged 18 years or older) to collect data necessary to inform the design of the peer-led HIVST intervention. Participants were selected in such a way as to represent different interest groups in each community (e.g. people engaged in fishing or fishing-related activities, farmers, *boda-boda* cyclists [groups of motorcycle riders], people engaged in business or other related commercial activities), with support from local leaders and village health teams (groups of trained local community members who provide basic health services at village level). During the FGDs, participants were asked about their perceptions towards HIVST in general and peer-led HIVST in particular; existing social groups or networks in the community and, if we were to distribute HIV self-kits to members of identified social groups, how best we could do that. Social networks were defined as loosely interconnected groups of people who engage in the same activity, or live together, or who associate for work or other reasons.

Focus group discussions were audio-recorded and transcribed verbatim by two trained Social Scientists with experience in the conduct of qualitative interviews. Data analysis was done manually following a thematic framework approach [26]. Initially, JN reviewed the scripts following *a priori* themes (perceptions of HIVST; anticipated fears about HIVST; potential acceptability of a peer-led HIVST program; qualities of a community-based HIVST distributor; and linkage to HIV care among those testing HIV-positive), and generated a matrix comprising representative quotations that supported each theme. JKBM and JN reviewed the matrix and revised it to remain with rich quotations that best supported each theme. The identified quotations were then organized by type of participant (i.e. young men, young women, or adult men), as shown in Table 1 below. Table 1 shows the themes, examples of quotations, and how these were used to inform the design and implementation of the peer-led HIVST intervention.

At least 21 social network groups were identified in the three study communities, including fishermen, boat pushers, *boda-boda* cyclists, footballers, netballers, farmers, sex workers, DREAMS groups, and savings/cash-round groups. Fishermen, sex workers, savings groups and DREAMS had the highest membership with between 100–500 members. DREAMS (Determined, Resilient, Empowered, AIDS-free, Mentored and Safe) is an ambitious, PEP-FAR-funded, public-private partnership that aims to reduce rates of HIV among adolescent girls and young women (AGYW) in the highest HIV burden countries, including Uganda. Since Rakai district is one of the districts targeted by the DREAMS Project in Uganda, members identified girls enrolled into this project as a specific group. Cross-membership was possible; that is, *boda-boda* cyclists could belong to a savings group that also included footballers. Netballers could also belong to DREAMS, and some girls could be sex workers. However, members indicated royalty to specific groups, for instance, *boda-boda* cyclists who belonged to savings group considered themselves more as *boda-boda* cyclists and girls enrolled in the DREAMS Project preferred to refer to themselves as the DREAMS group even if they belonged to other groups. This observation was noted among other groups.

**Table 1.** Focus group themes and quotes that informed the design of the intervention.

| Theme | Example quotes | Application to intervention design and implementation |
|---|---|---|
| **Acceptability of HIV self-testing** | *Young man*: "According to me, the way I see myself and my fellow young people, they will be willing to use HIV self-test kits because in most cases they are busy working so they do not get time to go to health centers for HIV testing and usually health centers are far requiring around UGX 2,000–4,000 (~US$0.5–1) for transport and yet I can spend a month without getting that money so if we are given those kits, it will help us to know our HIV status." | We used these findings to emphasize, during the training of peer-leaders, that HIV self-testing is easy to do, enhances confidentiality of test results, is convenient (given that one can perform the test in private) and reduces the burden of transport costs to go for HIV testing at health facilities. |
| | *Young woman*: "It's very good because your results will be private. If I test and find that I am HIV positive, it will remain between me and the health worker who I am going to see, no one else will know." | |
| | *Adult man*: "I have ever had an opportunity to see the HIV self-test kit and I moved with it showing it to people in this community and about 50 people liked it. Therefore, HIV self-test kits are user friendly, so many people will be willing to use them and self-test for HIV." | |
| **Fears about HIV self-testing** | *Young man*: "My concern will be on counselling, if a person has done an HIV self-test and got positive results alone without any kind counselling to prepare this person in receiving his own results, how will the anxiety be managed?" | We used these findings to train peer-leaders in basic counselling and referral skills. We identified Liaison Nurse Counsellors at the health facilities and included their telephone contacts in the consent forms. Respondents were requested to contact these Nurses any time before, during and after the test. We emphasized to the peer-leaders that in HIV self-testing, a person receives adequate information to enable them to make a decision to test for HIV but no face-to-face counselling is provided. |
| | *Young woman*: "When a person self-tests, he/she won't take the responsibility of going to the health center to get medicine because he/she doesn't have anyone to encourage him/her. He/she might be afraid of going for the medicine because he/she is afraid of swallowing it. . ." | |
| | *Adult man*: "My fear is it's very possible for someone to test HIV positive and doesn't go to the health facility to seek further HIV care especially if this person thought that he/she is HIV negative and the HIV self-test results show positive. Because he/she was not counseled enough, he/she may be reluctant going to the health facility to seek HIV confirmatory testing and HIV care." | |
| **Support needed to perform HIV self-testing** | *Young man*: "Before you give him/her that kit, you should educate him/her on how he/she is going to use it to test. If he/she tests positive, you encourage him/her to go to the health center to get treatment. If he/she tests negative, you encourage him/her to protect her/himself that if he/she is going to have sex, he/she should use a condom" | We trained the peer-leaders in all the necessary HIV self-testing processes, including what to do before, during and after HIV self-testing. Each peer-leader received a kit during the training and they used this kit for the practical sessions. We emphasized the need for peer-leaders to train their social network members about how to perform the test before giving them the kits. |
| | *Young woman*: "He/she [HIVST kits user] needs counselling on how he/she will use it. Maybe they can tell him/her what to do if he/she tests HIV positive." | |
| | *Adult man*: "The support would be training about how to use self-test kits because if people are trained on how to use these kits then they will use them with them with ease." | |
| **Potential acceptability of peer-led HIV self-testing** | *Adult man*: "People will take up the program because it is private, since going to heath facilities makes them fear that they will be seen by other people. This method is very confidential it's upon the individual himself to self-test for HIV and keep the results to himself or tell the peer leader if she/he wants." | This was important for the design of the intervention; it ensured that the program would be accepted by community residents. We used these findings to emphasize the benefits of such a program, including the fact that one can easily obtain a kit from a peer without going to the health facilities. |
| | *Young man*: "The program is good and we shall embrace it, people will be able to self-test for HIV instead of going to long queues in public health facilities where there is no privacy." | |
| | *Young woman*: "They [people in the community] will feel comfortable because they live near the person who is going to give them kits. They will not need transport to go to another place." | |

(*Continued*)

**Table 1.** (Continued)

| Theme | Example quotes | Application to intervention design and implementation |
|---|---|---|
| **Qualities of a community-based HIV self-test kits distributor** | *Young woman*: "He/she should be friendly with everyone, trustworthy, keeps secrets, a resident who can be accessed any time. He/she should be educated and enlightened. He/she should be able to counsel others." | We used these findings to define the qualities of a peer-leader. This information was useful during community meetings that were convened for members to select their peer-leaders. |
| | *Young man*: "He/she must be someone who will be able to maintain confidentiality." | |
| | *Adult man*: "…They should not be mobile workers; they should be easily found within the community and can easily find the people they want to distribute kits to." | |
| **HIV self-test kits distribution within social network groups** | *Adult man*: "I suggest that we get a leader from each group and give him the responsibility of distributing HIV self-test kits to the rest of the group members." | These findings helped to emphasize the need to identify a leader within each group to handle the distribution of HIV self-test kits to members in his/her group. During the distribution of kits, we asked peer-leaders to assign chits to those recommended for study enrolment. We asked self-testers to return used kits to the Liaison Nurse Counsellor after self-testing. |
| | *Young man*: "Let me [give] an example, we [are] here in this group that has a name, we need slips so that you give out a kit and a paper and after testing, you return the kit to the health worker. That kit should have that person's name… those slips are going to help us note the number of kits that we have given out and the number that has been returned. We will know how many people have been able to know their status." | |
| | *Young woman*: "There's someone who heads the DREAMS program, it necessitates training that person because they can easily get to her since she is always around. They should train her so that she is the one that you meet and give the kits to." | |
| **How to enhance linkage to HIV care among those testing HIV-positive** | *Young woman*: "Those who are given kits to distribute should counsel the people who go and get kits from them. They should tell them that if you test HIV positive, you should go to the health centre. But if you give it to him/her without counselling him/her, he/she can see that she/he is HIV positive and stay back and fail to go to the health centre to get medicine." | We used these findings to emphasize, during the training of peer-leaders, the need to encourage their social network members to seek confirmatory HIV testing at the designated health facilities, and if confirmed to be HIV-positive, to enrol into HIV care as required. However, we did not ask peer-leaders to take HIV drugs to their members. |
| | *Adult man*: "A peer educator can intervene and take HIV treatment to the homes of the self-tested [HIV-positive] individuals." | |
| | *Young man*: "… the peer leaders distributing HIV self-test kits should calmly counsel whoever they distribute the kit [to] and tell them that in case they self-test HIV positive, they should go to the health facility and seek HIV care." | |

**Peer-leader selection, training and selection of social network members.** In June 2019, using information from the FGDs (see Table 1 above), we convened meetings in the community in which we asked community residents to select one peer-leader per social network grouping, for a total of 34 peer-leaders (11 young females aged 15–24 years; 10 young men aged 15–24 years and 13 adult men aged 25+ years). The selection process was guided by pre-set selection criteria, e.g., one had to know how to read and write; be a permanent resident of the area who is available in the community for most of the time; be known in the group in which they were selected; be known to keep people's secrets and be approachable. Peer-leaders received a three-day training on oral HIV self-testing processes; basic counselling, communication and referral skills; and how to approach social network members at the time of distributing the kits. They also received a practical demonstration of how the HIV self-testing exercise is conducted, including watching a video clip audio-translated into *Luganda*, the predominant local language. Each peer-leader received one kit for their own practical use, if needed. At the end of the training, each peer-leader was asked to nominate up to 20 members within their social networks. After the nomination, peer-leaders met in groups to identify members that could have been selected by more than one peer-leader (due to cross-

membership in the groups) and decided to nominate each member for only once. Research has demonstrated that 20 members reliably captures variability for most network characteristics [27]. Nominated individuals had to be young people (15–24 years, both male and female) or adult men (25+ years) personally known to the peer-leader. Since adolescent boys and young men were already included in the category of 'young people', we decided to target adult men starting from age 25 upwards. Nominated individuals had to be HIV-negative or of unknown status at the time of nomination; should not have tested for HIV in the past three months preceding nomination, and if below 18 years, they had to be an emancipated minor. An emancipated minor was defined as a young man or woman aged below 18 years of age who was living on their own (i.e. renting their own room, away from their parents) or married at the time of the study. Peer-leaders were informed that their nominated social network members would be screened for eligibility and only those found to be eligible would be enrolled into the study.

**Distribution of HIV self-test kits by peer-leaders.** In July 2019, a total of 298 individuals who were found to be eligible for the intervention were administered a baseline questionnaire (see '*data collection procedures and methods*' below) and thereafter requested to contact their peer-leader (i.e. the peer-leader who nominated them) to receive their HIV self-test kits. Peer-leaders distributed OraQuick Advance Rapid HIV-1/2 antibody test kits (OraSure Technologies) packaged with instructional materials in English and *Luganda*, the main local language used in the area. These instructional materials were specially designed for the OraQuick Advance Rapid HIV-1/2 antibody test. Each peer-leader received the number of kits equivalent to the number of his/her social network members who had been found to be eligible to participate in the intervention. To facilitate the HIV self-test kits distribution process, we gave each peer-leader a list of 'pre-qualified' social network members who had been found eligible to participate in the intervention to rule out the possibility of giving out kits to individuals who did not qualify to receive them. Before distributing the kits to their social network members, peer-leaders were instructed to demonstrate how the HIV self-testing exercise is conducted, including how to open the HIV self-kit packet; how to obtain the kit from the packet; how to obtain the oral swab from the mouth; how to place the test kit into the buffer solution; how to time the 20 minutes needed for the kit to show the results; and how to read the results. Peer-leaders were instructed to emphasize that testers should not eat anything or brush their teeth at least 30 minutes to the test and that they should conduct the test in a well-lit place in order to be able to read their results clearly. Finally, peer-leaders were requested to inform their social network members to return used (or unused) kits to the study team or to a Liaison Nurse at their nearest health facility after performing the test and reading their own results. The HIVST distribution process took approximately one month. Peer-leaders received up to $4 to meet their travel and other incidental costs (depending on distances covered) during the distribution of HIVST kits while social network members received $1 as travel refund to return used (or unused) kits to the designated health facility. Respondents received up to $4 at each visit as travel refund (based on distance travelled) and compensation for time taken to respond to survey questions.

## Data collection procedures and methods

Both baseline (July 2019) and follow-up data (August 2019) were collected using paper-based questionnaires administered to study participants by same-sex interviewers. Baseline data were collected prior to distribution of HIVST kits while follow-up data were collected after the distribution exercise had been completed. Baseline data were collected on socio-demographic and behavioural characteristics, HIV testing history, whether or not respondents had ever

heard of HIV self-testing, willingness to self-test, kind of support that respondents would need if they were to self-test, HIV self-testing preferences (e.g. where they would prefer to pick the HIV self-test kits from within their community). Follow-up data were collected from 282 respondents. Follow-up data were collected on HIV self-testing experiences since baseline, whether or not users experienced any adverse events, and confirmatory HIV testing among all users, and linkage to HIV care among those that were confirmed to be HIV-positive. Individuals who were confirmed to be HIV-positive were enrolled into HIV care at their nearest health facility following the government of Uganda's Test and Treat Policy [28].

## Measures

The two primary outcomes of interest were: a) feasibility of the intervention and b) acceptability of HIV self-test kits distributed by peer-leaders to social network members. The secondary outcomes were: a) proportion of first-time HIV-positive individuals identified; and b) proportion of first-time HIV-positive individuals linked to HIV care. **Feasibility** was measured as a function of the peer-leaders' ability to successfully distribute HIVST kits to members of their social networks. The intervention was deemed feasible if peer-leaders distributed up to 70% of the kits they received and if >80% of the social network members returned used/unused kits to health facility. **Acceptability** was defined from the HIVST kit recipient's point of view as the percentage of social network members who received and used the kit to self-test for HIV. Peer-led HIVST was deemed acceptable if >80% of social network members accepted to receive the kits from their trained peer-leaders, and if >80% of those that received kits from peer-leaders used them to self-test for HIV.

## Data analysis

As already noted, qualitative data were analysed manually, following a thematic framework approach. Quantitative data were entered, double-entered and validated using EpiData software (version 3.1, EpiData Association, Odense, Denmark). Data were later exported to STATA (version 13) for analysis. We summarized continuous data using medians with inter-quartile ranges, and categorical data using proportions. Chi Square tests were conducted to compute statistical differences between selected groups.

## Ethical considerations

This study was reviewed and approved by the Makerere University School of Public Health's Higher Degrees, Research and Ethics Committee (Protocol #: 649) and the Yale University School of Public Health Institutional Review Board (Protocol #: 2000024945) and cleared by the Uganda National Council for Science and Technology. We obtained written informed consent from all the respondents. Individuals aged 15–17 years of age were enrolled only if they were emancipated minors. Emancipated minors were considered to be eligible to provide their own consent without the need for parental/guardian consent, as per guidance from the Uganda National Council for Science and Technology (UNCST) [29].

## Results

### Respondents' socio-demographic and behavioural characteristics

A total of 653 social network members were screened for eligibility; of these, 305 (46.7%) were found to be eligible for study enrolment. These respondents constituted 87.9% of the expected 340 respondents. Reasons for ineligibility included: not being an emancipated minor (among those aged below 18 years), being below 15 years (among those aged 15–24 years) or above 24

**Table 2. Socio-demographic and behavioural characteristics of respondents at baseline.**

| Characteristics | Young Men (N = 71) (n, %) | Young Women (N = 97) (n, %) | Adult Men (N = 130) (n, %) | Total (N = 298) (n, %) |
|---|---|---|---|---|
| **Age (median, IQR)** | 21 (20–24) | 21 (20–23) | 35 (29–42) | 24 (21–33) |
| **Ever attended school** | | | | |
| *Yes* | 68 (95.8) | 96 (99.0) | 120 (92.3) | 284 (95.3) |
| *No* | 3 (4.2) | 1 (1.0) | 10 (7.7) | 14 (4.7) |
| **Ability to read and write in own local language** | | | | |
| *Yes* | 56 (82.4) | 79 (82.3) | 100 (76.9) | 235 (82.7) |
| *No* | 12 (17.7) | 17 (17.7) | 20 (15.4) | 49 (17.2) |
| **Marital status** | | | | |
| *Never married* | 28 (39.4) | 17 (17.5) | 11 (8.5) | 56 (18.8) |
| *In a relationship but not married* | 26 (36.6) | 25 (25.8) | 25 (19.2) | 76 (25.5) |
| *Married/long-term union* | 15 (21.1) | 48 (49.5) | 84 (64.6) | 147 (49.3) |
| *Divorced/Separated* | 2 (2.8) | 4 (4.1) | 10 (7.7) | 16 (5.4) |
| *Widowed* | 0 (0.0) | 3 (3.1) | 0 (0.0) | 3 (1.0) |
| **Main occupation** | | | | |
| *Fishing* | 14 (19.7) | 0 (0.0) | 20 (15.4) | 34 (11.4) |
| *Fishing-related activity* | 13 (18.3) | 1 (1.0) | 15 (11.5) | 29 (9.7) |
| *Sex worker* | 0 (0.0) | 10 (10.3) | 0 (0.0) | 10 (3.4) |
| *Peasant farmer* | 9 (12.7) | 17 (17.6) | 43 (33.1) | 69 (23.2) |
| *Salaried* | 2 (2.8) | 10 (10.3) | 10 (7.7) | 22 (7.4) |
| *Business/Commercial* | 14 (19.7) | 16 (16.5) | 27 (20.8) | 57 (19.1) |
| *Casual worker* | 5 (7.0) | 16 (16.5) | 5 (3.8) | 26 (8.7) |
| *House wife* | 0 (0.0) | 8 (8.2) | 0 (0.0) | 8 (2.7) |
| *Pupil/Student/No occupation* | 7 (9.9) | 10 (10.3) | 0 (0.0) | 17 (5.7) |
| *Self Employed* | 7 (9.9) | 9 (9.3) | 10 (7.7) | 26 (8.7) |
| **Ownership of a mobile phone** | | | | |
| *Yes* | 64 (90.1) | 61 (62.9) | 103 (79.2) | 228 (76.5) |
| *No* | 7 (9.9) | 36 (37.1) | 27 (20.8) | 70 (23.5) |
| **Number of sexual partners last had sex with in the past three months** | | | | |
| *None* | 2 (2.8) | 2 (2.1) | 1 (0.8) | 5 (1.7) |
| *1 sexual partner* | 29 (40.9) | 50 (51.6) | 69 (53.1) | 148 (49.7) |
| *2 sexual partners* | 9 (12.7) | 12 (12.4) | 24 (18.5) | 45 (15.1) |
| *3+ sexual partners* | 31 (43.7) | 33 (34.0) | 36 (27.7) | 100 (33.6) |
| **Ever tested for HIV and received results** | | | | |
| *Yes* | 61 (85.9) | 89 (91.7) | 124 (95.4) | 274 (92.0) |
| *No* | 10 (14.1) | 8 (8.3) | 6 (4.6) | 24 (8.0) |

years (among young women); being HIV-positive; and having tested for HIV within the past three months. Of 305 eligible respondents, 298 (97.7%) turned up for the baseline interview while seven did not. Table 2 shows the socio-demographic and behavioural characteristics of the respondents that were interviewed at baseline.

Of the 298 respondents, 56.4% ($n$ = 168) were young people (15–24 years) and 43.6% ($n$ = 130) were adult men aged 25+ years. The median age was 21 (interquartile range [IQR]: 20–24) and 35 years (IQR: 29–42) among young people and adult men, respectively. Ninety-five per cent of the respondents ($n$ = 284) had ever attended school; of these, 82.7% ($n$ = 235) were able to read and write in their local language. Nearly half of the respondents (49.3%,

*n* = 147) were married or in stable long-term union while 25.5% (*n* = 76) were in a relationship but not married. Nearly a quarter of the respondents (23.2%, *n* = 69) reported that they were engaged in peasant agriculture; 21.1% (*n* = 63) were engaged in fishing or fishing-related activities (e.g. net repairing, boat-making, etc.), while 19.1% (*n* = 57) were engaged in business or other commercial activities. A higher proportion of young men (38.0%, *n* = 27) than adult men (26.9%, *n* = 35) were engaged in fishing or fishing-related activities while 10.5% (*n* = 10) of young women reported that they were engaged in sex work.

Slightly more than three-quarters (*n* = 228) of the respondents owned a mobile phone, with mobile phone ownership much higher among young men (90.1%, *n* = 64) than young women (62.9%, *n* = 61). When asked about the number of sexual partners that they had had sex with in the three months preceding the survey, nearly half of the respondents (49.7%, *n* = 148) reported that they had sex with only one sexual partner while 33.6% (*n* = 100) reported that they had sex with 3+ sexual partners during this period. A higher proportion of young men (43.7%, *n* = 31) reported engaging in sex with 3+ partners than their female counterparts (34%, *n* = 33). Ninety-two per cent (*n* = 274) of the respondents had ever tested and received their HIV test results prior to the survey.

## Ever heard of HIVST and willingness to self-test for HIV

Table 3 shows the proportion of respondents that had ever heard of oral HIV self-testing (HIVST) and the proportion of respondents that were willing to self-test at baseline. Overall, 69.1% (*n* = 206) of the respondents had ever heard of oral HIVST with a higher proportion of young women (74.2%, *n* = 72) than young men (60.6%, *n* = 43) reporting that they had ever heard of oral HIVST. Most respondents indicated that they received information about HIVST from community health volunteers or the health facility within their area. All respondents indicated that they would be willing to use HIV self-test kits if they became freely available. When asked about what kind of support they would need in order to effectively perform the HIVST exercise, most respondents (84.9%, *n* = 253) indicated a need for pre- and post-test counselling; where to seek HIV care if they tested HIV-positive (78.6%, *n* = 235), and how to

**Table 3. Ever heard of HIV self-testing and willingness to self-test for HIV.**

| Characteristic | Young Men (N = 71) (n, %) | Young Women (N = 97) (n, %) | Adult Men (N = 130) (n, %) | Total (N = 298) (n, %) |
|---|---|---|---|---|
| **Ever heard of oral HIV self-testing** | | | | |
| *Yes* | 43 (60.6) | 72 (74.2) | 91 (70.0) | 206 (69.1) |
| *No* | 28 (39.4) | 25 (25.8) | 39 (30.0) | 92 (30.9) |
| **Would you be willing to use a kit to self-test for HIV if kits were freely available?** | | | | |
| *Yes* | 71 (100.0) | 97 (100.0) | 130 (100.0) | 298 (100.0) |
| *No* | 0 (0.0) | 0 (0.0) | 0 (0.0) | 0 (0.0) |
| **What kind of support would you need to perform HIVST if kits became available?** | | | | |
| *How to obtain the oral swab* | 38 (53.5) | 45 (46.4) | 72 (55.4) | 155 (52.0) |
| *How to perform the test itself* | 38 (53.5) | 47 (48.5) | 72 (55.4) | 157 (52.7) |
| *How to read the results* | 39 (54.9) | 50 (51.5) | 70 (53.8) | 159 (53.4) |
| *How to interpret the results* | 39 (54.9) | 51 (52.6) | 72 (55.4) | 162 (54.4) |
| *Pre-and post-testing counselling* | 58 (81.7) | 85 (87.6) | 110 (84.6) | 253 (84.9) |
| *Referral for HIV care if HIV-positive* | 59 (83.1) | 70 (72.2) | 106 (81.5) | 235 (78.6) |
| *How to dispose of the kit after use* | 49 (69.0) | 60 (61.9) | 82 (63.1) | 191 (64.1) |

dispose of used HIV self-test kits (64.1%, $n$ = 191). Other forms of support that respondents needed included: how to read (53.4%, $n$ = 159) and interpret the HIV test results (54.4%, $n$ = 162); how to perform the HIVST exercise (52.7%, $n$ = 157) and how to obtain the oral swab from the mouth (52.0%, $n$ = 155). Although a slightly higher proportion of young men than women reported that they needed support in how to obtain the oral swab, how to perform the test itself, and how to read and interpret results than young women, these differences were not statistically significant ($P$>0.05).

## HIV self-testing preferences

Table 4 shows the different HIVST preferences of the respondents at baseline. There was a higher preference for unsupervised than supervised HIVST (61.1%, $n$ = 182 vs. 38.9%, $n$ = 116) and this was true for all categories of respondents. Virtually all respondents (99.3%, $n$ = 296) reported that they would be willing to receive HIVST kits from a trained local person in their community. When asked which kind of person that they would prefer to receive HIVST kits from, 70.8% ($n$ = 211) preferred a trained community health volunteer while 48% ($n$ = 143) preferred a friend or relative. When asked what qualities the community-based HIVST distributor should have, 51.7% ($n$ = 154) preferred someone who could keep secrets; 49.0% ($n$ = 146) preferred someone who is approachable; while 37.9% ($n$ = 113) preferred someone who could read and write. Seventy-six per cent ($n$ = 227) of the respondents reported that they did not mind the sex of the HIVST distributor. When asked where they would prefer to pick the kits from, 70.8% ($n$ = 211) preferred to collect the kits from the home of a local HIVST distributor; 59.7% ($n$ = 178) preferred to have the kits delivered to their own homes while 46.6% ($n$ = 139) preferred to pick the kits from the nearest health facility. A lower proportion of young men (40.8%, n = 29) and adult men (42.3%, n = 55) preferred to collect kits from the health facility than young women (56.7%, n = 55). Ninety-nine percent of the respondents ($n$ = 295) indicated that they would be willing to initiate HIV treatment immediately if they were found to be HIV-positive.

## Preliminary intervention effects

A total of 298 kits were given to 34 peer-leaders to distribute to 298 eligible social network members (one kit per member). Based on our process data, 99.3% ($n$ = 296) of the kits were distributed to social network members, as expected, with only two kits remaining undistributed. A majority of the peer-leaders (82.3%, $n$ = 28) distributed between 8–10 kits each. Of the 282 respondents (94.6% of baseline) that were interviewed at follow-up, 98.2% ($n$ = 277) reported that they used the HIVST kits to self-test for HIV. Of these, 93.1% ($n$ = 258) reported that they performed the test themselves. Virtually all respondents (99.6%, $n$ = 281) reported that they would recommend the kits to other people, including their close friends; 98.9% (n = 278) reported that they would recommend that HIVST kits continue to be distributed in the community by trained local distributors. Table 5 shows the preliminary intervention effects based on follow-up data. Twenty-one (7.4%) respondents tested HIV-positive: 12 (57.1%) were first-time testers while 9 (42.9%) were repeat HIV-positive testers.

All 12 first-time testers reported that they sought confirmatory HIV testing at the designated health facility. Of these, 5 (41.7%) respondents did so on the same day of self-testing while 7 (58.3%) did so after the first day but within the first week of HIV self-testing. Confirmatory HIV test results showed that 2 (16.7%) respondents tested HIV-negative while 10 (83.3%) were confirmed as HIV-positive. When asked if they were linked to HIV care as per Uganda government's HIV test and treat policy, 9 (90%) of the 10 confirmed HIV-positive individuals reported that they were linked to HIV care at the same health facility where they

Table 4. HIV self-testing preferences.

| Characteristics | Young Men (n = 71) (n, %) | Young Women (n = 97) (n, %) | Adult Men (n = 130) (n, %) | Total (n = 298) (n, %) |
|---|---|---|---|---|
| **Would you prefer supervised or unsupervised HIV self-testing?** | | | | |
| *Peer-leader supervised HIV self-testing* | 28 (39.4) | 33 (34.0) | 55 (42.3) | 116 (38.9) |
| *Unsupervised HIV self-testing* | 43 (60.6) | 64 (66.0) | 75 (57.7) | 182 (61.1) |
| **Would you be willing to receive kits from a trained local person in your community?** | | | | |
| *Yes* | 71 (100.0) | 96 (99.0) | 129 (99.2) | 296 (99.3) |
| *No* | 0 (0.0) | 1 (1.0) | 1 (0.8) | 2 (0.7) |
| **If HIV self-test kits became available, which kind of person would you like to distribute them in the community?** | | | | |
| *Trained community health volunteer* | 48 (67.6) | 72 (74.2) | 91 (70.0) | 211 (70.8) |
| *Friend/relative* | 33 (46.4) | 48 (49.5) | 62 (47.7) | 143 (48.0) |
| *Sexual partner* | 8 (11.3) | 20 (20.6) | 10 (7.7) | 38 (12.8) |
| *Local council official* | 23 (32.3) | 26 (26.8) | 29 (22.3) | 78 (26.2) |
| *Religious official* | 5 (7.0) | 13 (13.4) | 13 (10.0) | 31 (10.4) |
| **Preferred qualities of the HIV self-test kits distributor** | | | | |
| *Someone who can keep secrets* | 29 (40.8) | 59 (60.8) | 66 (50.8) | 154 (51.7) |
| *Someone who can read and write* | 18 (25.3) | 43 (44.3) | 52 (40.0) | 113 (37.9) |
| *Someone who has ever tested for HIV* | 13 (18.3) | 32 (33.0) | 40 (30.8) | 85 (28.5) |
| *Someone who is approachable* | 36 (50.7) | 50 (51.5) | 60 (46.1) | 146 (49.0) |
| *Someone who is available at all times* | 26 (36.6) | 32 (33.0) | 42 (32.3) | 100 (33.6) |
| **Would you mind if the HIV self-test kits distributor was of the opposite sex?** | | | | |
| *Yes, I would mind* | 19 (26.8) | 33 (34.0) | 19 (14.6) | 71 (23.8) |
| *No, I would not mind* | 52 (73.2) | 64 (66.0) | 111 (85.4) | 227 (76.2) |
| **If HIV self-test kits became available in this community, where would you like to receive them from?** | | | | |
| *Own home* | 37 (52.1) | 62 (63.9) | 78 (60.0) | 178 (59.7) |
| *Own work-place* | 16 (22.5) | 27 (27.8) | 23 (17.7) | 66 (22.1) |
| *Local distributor's work place* | 8 (11.3) | 32 (32.9) | 31 (23.8) | 71 (23.8) |
| *Anywhere in community, not at home* | 6 (8.5) | 4 (4.1) | 6 (4.6) | 16 (5.4) |
| *Home of local HIV self-test kit distributor* | 46 (64.8) | 70 (72.2) | 95 (73.1) | 211 (70.8) |
| *Health facility* | 29 (40.8) | 55 (56.7) | 55 (42.3) | 139 (46.6) |
| *Drug shop* | 9 (12.7) | 23 (23.7) | 19 (19.6) | 51 (17.1) |
| **If you self-tested HIV-positive, would you be willing to be initiated on ART immediately?** | | | | |
| *Yes* | 71 (100.0) | 97 (100.0) | 127 (97.7) | 295 (99.0) |
| *No* | 0 (0.0) | 0 (0.0) | 3 (2.3) | 3 (1.0) |

sought confirmatory HIV testing. All nine were linked to HIV care within the first week of HIV self-testing.

## Discussion

Our study of the feasibility and acceptability of a peer-led HIV self-testing intervention found high levels of feasibility (99.3% of the kits given to the peer-leaders were distributed) and acceptability (96.6% of the distributed kits were used by the respondents) associated with peer-led HIV self-testing. The onus of this intervention was in the use of local people, who were members of existing social networks, who were trained in how to distribute HIVST kits and

**Table 5. Preliminary intervention effects.**

| Characteristic | N | Frequency |
| --- | --- | --- |
| | | n(%) |
| **Overall test results** | 282 | |
| *Positive* | | 21 (7.4) |
| *Negative* | | 258 (91.5) |
| *vIndeterminate* | | 3 (1.1) |
| *Don't know/don't remember* | | 0 (0.0) |
| **Was this your first time to test and found yourself positive or you knew about it?** | 21 | |
| *First-time HIV-positive result* | | 12 (57.1) |
| *Repeat HIV-positive result* | | 9 (42.9) |
| *Can't tell* | | 0 (0.0) |
| **Went for confirmatory HIV testing** | 12 | |
| *Yes* | | 12 (100.0) |
| *No* | | 0 (0.0) |
| **Confirmatory results** | 12 | |
| *Positive* | | 10 (83.3) |
| *Negative* | | 2 (16.7) |
| *Don't know/don't remember* | | 0 (0.0) |

how to educate their social network members in how to use the kits to self-test for HIV. Our baseline study shows that 99.3% of the respondents reported that they would be willing to receive HIVST kits obtained from trained local people in their community. Moreover, when the kits were actually availed through trained peer-leaders, almost all (98.2%) of those that received HIVST kits reported using them to self-test for HIV. Indeed, prior to receiving the kits, a high proportion of respondents (71%) preferred to receive HIVST kits from a trained community health volunteer; after using the kits, 98.9% reported that they would recommend that kits continue to be distributed through trained local community distributors. These findings suggest that the use of peer-leaders to distribute HIVST kits to members of their social networks is a feasible strategy that could potentially be scaled up to other fishing communities.

Study findings are in direct consonance with findings reported in other prior peer-based HIVST studies. In a pilot study conducted in four fishing communities along the shores of Lake Albert in Uganda, Choko et al. [30] reported that 81.9% of men that received HIVST kits from trained peers used them to self-test for HIV while Okoboi et al. [31] reported that 95% of men who have sex with men (MSM) who received HIVST kits through existing MSM peer networks self-tested for HIV in Kampala, Uganda. In a study conducted to assess the uptake of HIV testing among men and adolescents in Malawi, Zambia and Zimbabwe, Hatzold et al. [32] found that the use of lay people who were trained to distribute HIV self-test kits in the community resulted in high HIV testing rates among men and young people: nearly half of the HIV self-testers (48.2%) were men and up to 43% of testers were aged 16–24 years. Collectively, our results, together with previous findings on this subject, suggest that a peer-led HIV self-testing intervention is feasible and can improve HIV testing rates in populations that are normally missed through conventional HIV testing services, including the fisher-folk.

Our preliminary results suggest that peer-led HIVST identifies previously undiagnosed HIV infections: 57% of HIV-positive individuals had never tested for HIV. Similar findings have been reported in previous studies [30, 31]. Our results further confirm that peer-led HIVST can facilitate timely linkage to HIV care, with 90% of those who were confirmed as HIV-positive linking to HIV care within the first week of HIV self-testing. This finding is not surprising considering that 95% of the respondents interviewed at baseline reported that they

would be willing to link to HIV care if they were found to be HIV-positive. However, our finding of a high linkage to HIV care following HIV self-testing is much higher than previously reported in facility-based HIV self-testing studies [33, 34]. Several factors could have contributed to the high linkage to HIV care observed in our study: a) the use of community-based peer-leaders to distribute HIVST kits to fellow men could have motivated men to talk about HIV testing and the need for linkage to HIV care (if HIV-positive) with their male peer-leaders, as has been adduced from prior social network-based studies; [35, 36] b) the high coverage of HIV services, including antiretroviral therapy within the targeted fishing community [4] could have acted as a motivator for respondents to seek HIV services as a normative behavior; and c) the presence of two ART-accredited health facilities that served the three fishing communities, located within easy reach of the study population, could have presented more of a motivator than a barrier for respondents to access HIV services. However, given that young people and men usually do not prefer to go to health facilities for HIV and other health services, we believe that the additional encouragement by their peer-leaders could have helped them to overcome the laxity in using health facility-based HIV services. Nevertheless, since this study was not powered to detect factors associated with HIV testing uptake and linkage to HIV care, we cannot conclude with certainty if one or a combination of these factors contributed to the high linkage to HIV care observed in this setting. These observations call for further research on this subject to assess these and other factors associated with HIV self-testing and linkage to HIV care in other Ugandan fishing community settings.

This study had a number of limitations and strengths. As already noted, the study was conducted in a setting with high coverage of ART services; [9] this could have contributed to the high uptake of HIV testing and linkage to HIV care services, beyond what could be observed in a population with limited access to HIV services. In addition, while previous studies in the same setting have yielded HIV prevalence levels ranging between 37–41%, [4, 20, 37] only 7.4% of those studied were found to be HIV-positive; 43% of whom were repeat HIV-positive testers. So, it is likely that we targeted a low-risk population that was more motivated to test for HIV, since prior studies suggest that low-risk rather than high-risk individuals are more likely to test for HIV [38, 39]. Thus, our findings may not be generalizable to other residents of the fishing community or other high HIV prevalence fishing communities for that matter. Thirdly, our study was conducted in a setting where HIV testing coverage was already high [9]; therefore, the high HIV testing rates observed in the studied population could have been driven by the fact that people were already motivated to test for HIV [40] which may not be the case in a different setting with limited coverage of HIV services. Fourthly, we used one (1) peer-leader to reach every 8–10 participants which could have influenced HIV testing uptake in both positive and negative ways. Since we used a peer-leader who was known to their social network members, this peer-leader could have swayed HIV testing uptake rates upwards, thereby explaining the higher uptake rates reported in this paper. However, in the event that some network members did not feel comfortable to receive kits from their selected peer-leaders; this could have prevented some members from accepting the kits, resulting in lower testing rates. Our findings show that only two (2) individuals refused to take their kits, suggesting that this approach has the potential to increase HIV testing uptake especially in hard-to-reach populations. Besides, we believe that our study, which used trained local people (peer-leaders) to reach members of their social networks with HIV testing and linkage to HIV care services, is the first study to utilize this approach in a fishing community setting and the second one to target fishing communities with HIV self-testing in Uganda [30]. Thus, the above-mentioned study limitations notwithstanding, our peer-led HIV self-testing model demonstrates the feasibility and acceptability of similar interventions to reach highly mobile and hard-to-reach populations, including fisher-folk.

## Conclusion

Our study shows that a social network-based, peer-led HIVST intervention in a hyperendemic fishing community is highly feasible and acceptable, and achieves high linkage to HIV care among newly diagnosed HIV-positive individuals. These findings suggest that a peer-led HIV self-testing model can help to address barriers associated with HIV testing and linkage to HIV care among the fisher-folk.

## Supporting information

**S1 Data. Dataset used for analysis.**
(DTA)

**S1 Study tool. Baseline questionnaire (English).**
(DOC)

**S2 Study tool. Baseline questionnaire (Luganda).**
(DOC)

**S3 Study tool. Focus group discussion guide (English).**
(DOC)

**S4 Study tool. Focus group discussion guide (Luganda).**
(DOC)

## Acknowledgments

We acknowledge support from the Rakai Health Sciences Program in the implementation of this study, members of the research team that conducted the study, and study participants for participating in this study. Acknowledgement also goes to the peer-leaders without whom the successful implementation of the intervention reported in this paper would not have been possible. Finally, we acknowledge the support of health workers at the two ART-accredited health facilities for their support during confirmatory HIV testing and linkage to HIV care of individuals who sought these services from the health facilities.

## Author Contributions

**Conceptualization:** Joseph K. B. Matovu, Laura M. Bogart, Joseph Kagaayi, David Serwadda, Rhoda K. Wanyenze, Albert I. Ko, Ann E. Kurth.

**Formal analysis:** Joseph K. B. Matovu, Jennifer Nakabugo.

**Funding acquisition:** Joseph K. B. Matovu, Joseph Kagaayi, David Serwadda, Albert I. Ko, Ann E. Kurth.

**Methodology:** Joseph K. B. Matovu, Laura M. Bogart, Joseph Kagaayi, David Serwadda, Rhoda K. Wanyenze, Albert I. Ko, Ann E. Kurth.

**Project administration:** Joseph K. B. Matovu, Jennifer Nakabugo, Joseph Kagaayi.

**Supervision:** Joseph K. B. Matovu, Jennifer Nakabugo, Joseph Kagaayi, David Serwadda, Rhoda K. Wanyenze.

**Validation:** Joseph K. B. Matovu, Jennifer Nakabugo.

**Writing – original draft:** Joseph K. B. Matovu, Laura M. Bogart, David Serwadda, Rhoda K. Wanyenze, Albert I. Ko, Ann E. Kurth.

**Writing – review & editing:** Joseph K. B. Matovu, Laura M. Bogart, David Serwadda, Rhoda K. Wanyenze, Albert I. Ko, Ann E. Kurth.

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
