## [Decision Letter · Decision Letter 0]

5 Jun 2020

PONE-D-20-13241

Feasibility and acceptability of a pilot, peer-led HIV self-testing intervention in a hyperendemic fishing community in rural Uganda

PLOS ONE

Dear Dr. Matovu,

Thank you for submitting your manuscript to PLOS ONE. After careful consideration, we feel that it has merit but does not fully meet PLOS ONE’s publication criteria as it currently stands. Therefore, we invite you to submit a revised version of the manuscript that addresses the points raised during the review process.

We look forward to receiving your revised manuscript.

Kind regards,

Joel Msafiri Francis, MD, MS, PhD

Academic Editor

PLOS ONE

Journal Requirements:

2. Please provide additional details regarding participant consent for both the qualitative and questionnaire-based sections of your study. In the ethics statement in the Methods and online submission information, please ensure that you have specified (1) whether consent was informed and (2) what type you obtained (for instance, written or verbal, and if verbal, how it was documented and witnessed). If your study included minors, state whether you obtained consent from parents or guardians. If the need for consent was waived by the ethics committee, please include this information.

3. Please include additional information regarding the interview guide and questionnaire used in the study and ensure that you have provided sufficient details that others could replicate the analyses. For instance, if you developed a questionnaire as part of this study and it is not under a copyright more restrictive than CC-BY, please include a copy, in both the original language and English, as Supporting Information.

Reviewers' comments:

Reviewer's Responses to Questions

**Comments to the Author**

1. Is the manuscript technically sound, and do the data support the conclusions?

Reviewer #1: Yes

Reviewer #2: Yes

2. Has the statistical analysis been performed appropriately and rigorously? 

Reviewer #1: Yes

Reviewer #2: Yes

3. Have the authors made all data underlying the findings in their manuscript fully available?

Reviewer #1: Yes

Reviewer #2: Yes

4. Is the manuscript presented in an intelligible fashion and written in standard English?

Reviewer #1: Yes

Reviewer #2: Yes

5. Review Comments to the Author

Reviewer #1: Matovu et al. present research titled “Feasibility and acceptability of a pilot, peer-led HIV self testing intervention in a hyperendemic fishing community in rural Uganda” in which the authors developed and evaluated a peer-led HIV self testing intervention for young individuals (15-24 years) and adult men (25+ years) in Rakai. This is an interesting and well-written manuscript, and will make an important contribution to the HIV prevention literature.

Some minor concerns and suggestions regarding specific sections of this manuscript are outlined below:

ABSTRACT

Page 2, Line 34: The word “services” needs to be deleted from this sentence.

Page 2, Line 51: “HIV prevalence was 7.6% (n=21). Of the 57.1% (n=12) first time HIV-positives, 100% sought confirmatory HIV testing” – Please clarify the denominators for the estimates of 7.6% and 57.1%.

INTRODUCTION

Page 4, Line 112: How does the prevalence of 48% compare with the general population in Uganda?

Page 5, Line 122: Some justification is needed for why young women were included in this study given their substantially higher levels of HIV testing (62.7%) compared to young men (37.3%).

MATERIALS AND METHODS

Page 12, Line 257: Each peer-leader was asked to nominate up to 20 potential participants because research has demonstrated that a figure of 20 reliably captures variability for most network characteristics. Please provide an explanation of how a ratio of 1 peer-leader to 8-10 participants could have influenced the results.

Page 13, Line 276: Were new instructions developed for how to self-test using the OraQuick Advance Rapid HIV-1/2 Antibody Test or was the OraQuick In-Home HIV Test used?

Page 15, Line 316: “The intervention was deemed feasible if peer-leaders distributed up to 70% of the kits they received and if >80% of the social network members returned used/unused kits to health facility.” Why would returning unused kits reflect feasibility? “Acceptability was defined from the HIVST kit recipient’s point of view as the percentage of social network members who received and used the kit to self-test for HIV. This seems to overlap with feasibility. Please clarify these definitions.

RESULTS

Page 17, Table 2: More than 90% of the sample had been previously tested for HIV and received their results. The discussion of how this could have influenced the results needs to be strengthened.

Reviewer #2: Dear Authors,

I found this paper interesting .Below I summarize my main points, in the hope that you might nonetheless find them constructive.

Comments:

1. The introduction is missing the significance of the study, the gaps in the literature that the study is filling, who stand to benefit from this work, how is it pushing the streams of the literature, how can this results be replicated into policy actions based on its novelty

2. We know that any new innovation is likely to be accepted mainly driven by curiosity – the authors may want to explain more broadly on what might be driving factors of uptake of HIVST in this region

3. The study is susceptible to both social desirability bias and non-response bias considering that the data collection strategy queried sensitive and potentially stigmatizing information. For example, HIV status, linkage to ART ect. The authors may want to explain some possible measures that were taken to prevent self-reported biases.

In conclusion, I recommend a revision to factor in these 3 points comprehensively.

Best wishes,

Reviewer

6. PLOS authors have the option to publish the peer review history of their article (what does this mean?). If published, this will include your full peer review and any attached files.

Reviewer #1: No

Reviewer #2: No

---

## [Author Response · Author response to Decision Letter 0]

8 Jun 2020

June 8th, 2020

The Editor 

PLoS ONE

Dear Sir,

Re: RESPONSE TO COMMENTS RAISED ON MS#: PONE-D-20-13241

Please find enclosed our revised manuscript based on the comments from the Academic Editor and the peer-reviewers.

We are glad for the opportunity to revise the paper, which has improved clarity of the main message in the paper. We than the reviewers for their insightfulness and the guidance from the Academic Editor.

We look forward to our paper being published in your prestigious journal.

Regards,

Joseph KB Matovu, MHS, PhD

Corresponding Author

POINT-BY-POINT RESPONSE TO THE ACADEMIC EDITOR AND PEER REVIEWERS’ COMMENTS

ACADEMIC EDITOR’S COMMENTS

Journal Requirements:

Response: We have ensured that the paper is formatted as per the journal’s style as much as we can do. We will be glad to address any other aspects that the Editor fills still needs tightening up. 

2. Please provide additional details regarding participant consent for both the qualitative and questionnaire-based sections of your study. In the ethics statement in the Methods and online submission information, please ensure that you have specified (1) whether consent was informed and (2) what type you obtained (for instance, written or verbal, and if verbal, how it was documented and witnessed). If your study included minors, state whether you obtained consent from parents or guardians. If the need for consent was waived by the ethics committee, please include this information.

Response: We have updated the ethics statement to indicate that we obtained written informed consent from the respondents. The information on the consenting process for minors is also provided. See lines 348-349, page 16.

3. Please include additional information regarding the interview guide and questionnaire used in the study and ensure that you have provided sufficient details that others could replicate the analyses. For instance, if you developed a questionnaire as part of this study and it is not under a copyright more restrictive than CC-BY, please include a copy, in both the original language and English, as Supporting Information.

Response: We have provided copies of the interview guide and questionnaire in English as well as in the local language. See S2-S5, page 32, for details.

PEER REVIEWERS’ COMMENTS

Reviewer #1: 

Matovu et al. present research titled “Feasibility and acceptability of a pilot, peer-led HIV self testing intervention in a hyperendemic fishing community in rural Uganda” in which the authors developed and evaluated a peer-led HIV self testing intervention for young individuals (15-24 years) and adult men (25+ years) in Rakai. This is an interesting and well-written manuscript, and will make an important contribution to the HIV prevention literature.

Some minor concerns and suggestions regarding specific sections of this manuscript are outlined below:

ABSTRACT

Page 2, Line 34: The word “services” needs to be deleted from this sentence.

Response: The sentence refers to interventions needed to reach young people and adult men with HIV services (not: young people and adult men with HIV). For that reason, we have opted to retain the word ‘services’ in line 34.

Page 2, Line 51: “HIV prevalence was 7.6% (n=21). Of the 57.1% (n=12) first time HIV-positives, 100% sought confirmatory HIV testing” – Please clarify the denominators for the estimates of 7.6% and 57.1%.

Response: We have corrected the HIV prevalence estimate to 7.4% (21 of 282 participants followed up) as shown elsewhere in the paper. Of the 21 HIV-positive individuals, 12 were first-time positives (i.e. 12 of 21), see line 51.

INTRODUCTION

Page 4, Line 112: How does the prevalence of 48% compare with the general population in Uganda?

Response: We have edited the statement to include the comparison with the general population of Uganda (line 114, page 4)

Page 5, Line 122: Some justification is needed for why young women were included in this study given their substantially higher levels of HIV testing (62.7%) compared to young men (37.3%).

Response: While a prior study identified higher HIV testing uptake among females than males (lines 81-82), we included women in the study for three basic reasons: a) women continue to be at increased risk of HIV infection; so, they continue to be a population in need of continued targeting with HIV interventions; b) an HIV testing uptake of 63% among females as reported by Mafigiri et al. (2017) is still below the 1st 90% threshold; so, efforts are still needed to reach the remaining 37% with HIV testing services; and c) due to the high HIV incidence in the fishing communities, continued access to HIV testing services is crucial for early identification of first-time HIV-positive women who can then be linked to HIV care.

MATERIALS AND METHODS

Page 12, Line 257: Each peer-leader was asked to nominate up to 20 potential participants because research has demonstrated that a figure of 20 reliably captures variability for most network characteristics. Please provide an explanation of how a ratio of 1 peer-leader to 8-10 participants could have influenced the results.

Response: Using 1 peer-leader to reach 8-10 participants could have influenced the results in both positive and negative ways: since we used a peer-leader who was known to their social network members, this could have influenced HIV testing uptake positively, and hence explain the high HIV testing uptake rates reported in this paper. However, in the event that some network members did not feel comfortable to receive kits from their selected peer-leaders; this could have prevented some of them from taking the kits to test for HIV. However, since only two people refused to take the kits, we can infer that this approach has the potential to increase HIV testing uptake especially in populations that are not easy to reach with HIV services. We have included this perspective in the discussion section, see lines 510-517, pages 24-25.

Page 13, Line 276: Were new instructions developed for how to self-test using the OraQuick Advance Rapid HIV-1/2 Antibody Test or was the OraQuick In-Home HIV Test used?

Response: We used instructional materials developed by OraSure Technologies for the OraQuick Rapid HIV-1/2 Antibody Test. 

Page 15, Line 316: “The intervention was deemed feasible if peer-leaders distributed up to 70% of the kits they received and if >80% of the social network members returned used/unused kits to health facility.” Why would returning unused kits reflect feasibility? “Acceptability was defined from the HIVST kit recipient’s point of view as the percentage of social network members who received and used the kit to self-test for HIV. This seems to overlap with feasibility. Please clarify these definitions.

Response: We placed emphasis on return of used kits as part of feasibility because we wanted to put in place a mechanism to tell if people had actually used the kits to self-test for HIV. If we only based on self-reports from the users, this could have resulted in unreliable estimates since some people could report that they used the kits when, in fact, they did not. We don’t see any overlap between our definition of ‘feasibility’ and that of ‘acceptability’: feasibility was defined from the peer-leaders’ perspective as the ability to distribute kits to their social network members while acceptability was defined from the perspective of the user. For instance, if the peer-leaders were able to distribute 80% of the kits but only 10% used the kits, then, it would not be worth scaling up this intervention: the ultimate purpose is in the use of the kits but we also wanted to assess if use of peer-leaders was itself feasible. 

RESULTS

Page 17, Table 2: More than 90% of the sample had been previously tested for HIV and received their results. The discussion of how this could have influenced the results needs to be strengthened.

Response: We thank the reviewer for this observation. We have taken care of the fact that prior HIV testing rates were already high at the time we did the study. See lines 510-513, page 24.

Reviewer #2: 

I found this paper interesting .Below I summarize my main points, in the hope that you might nonetheless find them constructive.

Comments:

1. The introduction is missing the significance of the study, the gaps in the literature that the study is filling, who stand to benefit from this work, how is it pushing the streams of the literature, how can this results be replicated into policy actions based on its novelty

Response: We thank the reviewer for this comment. However, we don’t agree that the gaps in knowledge are not well spelt out in the introduction section. Specifically, for both young people and adult men, we mentioned the low HIV testing rates and the need for innovative approaches to improve HIV testing uptake (as well as linkage to HIV care among those identified as HIV-positive). The low uptake of HIV testing services among young people and adult men, which continue to bear the brunt of the HIV epidemic, is a major gap in the literature that requires urgent attention. Our study aimed to assess the feasibility and acceptability of an intervention that can help to improve HIV testing uptake and linkage to HIV care among young people and adult men, as presented in the paper. Within the ‘Discussion’ section, we presented our appreciation of the findings, highlighting their significance in the HIV testing discourse (e.g. see lines 460-462, page 22), who is likely to benefit from the results (e.g. see lines 473-475), and, by implication, how the results can influence linkage to HIV care, etc. We believe that a discussion of these aspects within the ‘Discussion’ section helps to address the issues raised by the reviewer in this comment.

2. We know that any new innovation is likely to be accepted mainly driven by curiosity – the authors may want to explain more broadly on what might be driving factors of uptake of HIVST in this region

Response: This is an important observation. We agree that the use of HIV self-test kits as an innovation in the HIV testing discourse, could have led to the high uptake rates observed in the study. However, we know, through our interactions with people in the study communities, that these kits were already available at the local health facilities but people did not go to pick them from there. We believe that what drove the results was the use of local trained persons who were able to reach fellow members in the community with HIV self-test kits. This aspect is well discussed throughout the ‘Discussion’ section.

3. The study is susceptible to both social desirability bias and non-response bias considering that the data collection strategy queried sensitive and potentially stigmatizing information. For example, HIV status, linkage to ART ect. The authors may want to explain some possible measures that were taken to prevent self-reported biases. 

Response: This is also another important observation. To ensure that we minimized any biases, we asked participants to return used kits to the study team and also ensured that the same interviewer per participant for both baseline and follow-up visits to increase trust. Asking respondents to return used kits was important given that some people could have reported that they used the kits when they did not. In addition, we asked participants to go to their nearest health facility (there are only 2 health facilities in the fishing community) for confirmatory HIV testing: only those that were confirmed as HIV-positive were linked to HIV care. So, we had mechanisms to verify the reported linkage to HIV care by checking with the health facility records.

---

## [Decision Letter · Decision Letter 1]

30 Jun 2020

Feasibility and acceptability of a pilot, peer-led HIV self-testing intervention in a hyperendemic fishing community in rural Uganda

PONE-D-20-13241R1

Dear Dr. Matovu,

We’re pleased to inform you that your manuscript has been judged scientifically suitable for publication and will be formally accepted for publication once it meets all outstanding technical requirements.

Kind regards,

Joel Msafiri Francis, MD, MS, PhD

Academic Editor

PLOS ONE

Additional Editor Comments (optional):

Reviewers' comments:

Reviewer's Responses to Questions

**Comments to the Author**

1. If the authors have adequately addressed your comments raised in a previous round of review and you feel that this manuscript is now acceptable for publication, you may indicate that here to bypass the “Comments to the Author” section, enter your conflict of interest statement in the “Confidential to Editor” section, and submit your "Accept" recommendation.

Reviewer #1: All comments have been addressed

Reviewer #2: All comments have been addressed

2. Is the manuscript technically sound, and do the data support the conclusions?

Reviewer #1: Yes

Reviewer #2: Yes

3. Has the statistical analysis been performed appropriately and rigorously? 

Reviewer #1: Yes

Reviewer #2: Yes

4. Have the authors made all data underlying the findings in their manuscript fully available?

Reviewer #1: Yes

Reviewer #2: Yes

5. Is the manuscript presented in an intelligible fashion and written in standard English?

Reviewer #1: Yes

Reviewer #2: Yes

6. Review Comments to the Author

Reviewer #1: (No Response)

Reviewer #2: I have no further comments on this paper, and happy to recommend it for publication. The authors have address all the pending issues.

Thanks

7. PLOS authors have the option to publish the peer review history of their article (what does this mean?). If published, this will include your full peer review and any attached files.

Reviewer #1: No

Reviewer #2: **Yes: **Cyprian M Mostert

---

## [Editor Report · Acceptance letter]

28 Jul 2020

PONE-D-20-13241R1 

Feasibility and acceptability of a pilot, peer-led HIV self-testing intervention in a hyperendemic fishing community in rural Uganda 

Dear Dr. Matovu:

I'm pleased to inform you that your manuscript has been deemed suitable for publication in PLOS ONE. Congratulations! Your manuscript is now with our production department. 

Kind regards, 

on behalf of

Dr. Joel Msafiri Francis 

Academic Editor

PLOS ONE